# DIFFUSION MODELS ARE FEW-SHOT LEARNERS FOR DENSE VISION TASKS

## ABSTRACT

The ability to adapt to new, unseen tasks with only a handful of training examples is a key factor behind the unprecedented success of language models. However, in computer vision, few-shot adaption has largely focused on adapting to new semantic categories or answering new visual questions. Adapting a model to dense vision tasks—depth estimation, surface normal estimation, semantic segmentation—has only been possible with large amounts of training data and with custom decoder heads, since the output spaces and the required information for different tasks vary widely. For instance, depth estimation requires understanding geometry, while semantic segmentation relies on semantic information. In this paper, we found that the diffusion prior can effectively adapt to various dense tasks, and based on this, we introduce an adaptation mechanism that exploits a pretrained diffusion model for 12 different dense vision tasks using only a few training examples. We found that even modifying a small number of input variables is sufficient to effectively adapt to new unseen tasks. Building on this, we further improved the performance on few-shot dense tasks. Our key insight is to reframe all dense prediction tasks into a codebook-conditioned classification problem, even for continuous outputs. Specifically, we learn two set of parameters: (1) concept embeddings that condition the diffusion model to encode task-specific representations in their attention masks; and (2) codebook embeddings that recombine discrete outputs to continuous ones. With this novel design, we achieve state-of-the-art results across 12 datasets for few shot learning.

## 1 INTRODUCTION

Language models are few-shot learners (Brown et al., 2020). To adapt to new unseen tasks, they require only a handful of training examples. This capability has enabled the related subfield of natural language processing to transition from designing task-specific objectives and architectures to task-agnostic ones. In computer vision, many semantic tasks, including categorization (Tseng et al., 2020), detection (Kang et al., 2019), and visual question answering (Alayrac et al., 2022), can be similarly reformulated such that their output spaces use language. With this reformulation, they can use language models as decoder heads and inherit their few-shot adaptation capabilities.

Developing few-shot adaptation for dense vision tasks—depth estimation, surface normal estimation, semantic segmentation, etc.—remains challenging. Multi-task generalists (Wang et al., 2023a; Yang et al., 2024; Ji et al., 2023) can integrate multiple dense tasks within one framework, but they cannot adapt to unseen tasks in a few-shot setting. Past research on few-shot settings for dense vision tasks has mainly focused on semantic segmentation (Johnander et al., 2022; Min et al., 2021; Hong et al., 2022). However, these methods are typically effective only for tasks with discrete semantic outputs and are not suitable for tasks like depth estimation, which have continuous outputs. Furthermore, dense tasks like depth estimation require the model to learn knowledge that is fundamentally different from that required for semantic segmentation, making it even more challenging for previous methods, which focus on semantic segmentation, to adapt to tasks like depth estimation. VTM (Kim et al., 2023) is the only known work dedicated to addressing general dense tasks in a few-shot setting. However, it relies on extensive pixel-level annotations for multitask learning during the pre-training phase, which not only hinders its scalability but also limits its ability to generalize effectively during the few-shot adaptation phase. Recent works like VPD (Zhao et al., 2023) have shown that pre-trained

diffusion models encode prior knowledge useful for various dense tasks. However, leveraging this prior typically requires a large amount of task-specific training data to adapt to a new task.

In this paper, we find that a pre-trained diffusion model can better adapt to different downstream dense tasks using few-shot samples, compared to VTM or other pre-trained models, even without optimizing internal model parameters but instead optimizing the input. Building on this, we further combine CLIP with the diffusion prior to achieve even better adaptation across 12 downstream dense vision tasks. First, we demonstrate that pre-trained diffusion models can adapt to different downstream tasks with just a few examples. To achieve this, we learn **task-specific concept embeddings** that query the diffusion model for meaningful information. Similar concept embeddings have been utilized by DETR (Carion et al., 2020) and MaskFormer (Cheng et al., 2021) to represent objectness. We extend this idea to show that continuous measurements can also be effectively represented using discrete concept embeddings. These concept embeddings consist of only a few hundred parameters and require very few examples to train. While similar approaches have been applied to depth estimation (e.g., Adabins (Bhat et al., 2021)), our work is distinct in that we demonstrate how this idea can unleash the potential of pre-trained diffusion models to adapt to a wide variety of unseen dense tasks. More importantly, unlike Adabins, which incorporates concept embeddings as part of the final output head, we optimize these concept embeddings as part of the model's input. This allows different dense vision tasks to share the same backbone architecture. This is akin to prompt tuning in NLP (Li & Liang, 2021), where a unified backbone can represent entirely new tasks through different inputs, eliminating the need for task-specific heads to adapt to new tasks. This approach has the potential to unlock in-context adaptation for dense vision tasks. We find that optimizing only these input variables allows pre-trained diffusion models to slightly outperform state-of-the-art methods.

Second, building on the above findings, we further reformulate all continuous dense prediction tasks into a **codebook-conditioned classification problem** to better leverage the diffusion prior and further improve performance on these tasks. Simply converting each task into a classification problem is insufficient. For instance, in depth estimation, the distribution of depth values in an indoor scene differs significantly from that of an outdoor scene. To address the varying per-scene distributions of potential output values, we propose a codebook-conditioned classification approach. Building on the strong baseline provided by the diffusion prior, we additionally utilize CLIP features from the image to attend over a learnable codebook, where the codebook values represent multiple possible distributions. These attended codebook elements are then used to recompose the classification output into a continuous value. This approach enables us to better leverage the diffusion prior for few-shot adaptation across diverse dense tasks.

Experiments across 12 dense vision tasks reveal that our mechanism can adapt diffusion models to new dense tasks with as few as 10 training examples per task. We compare our method against other recently proposed methods applicable to few-shot general dense prediction tasks, including VPD (Zhao et al., 2023) and VTM (Kim et al., 2023). These results are consistent across 10 Taskonomy (Zamir et al., 2018) and 2 NYUv2 (Silberman et al., 2012) datasets. Moreover, our representations outperform those from popular vision encoders, including CLIP (Radford et al., 2021) and DINOv2 (Oquab et al., 2023). These experimental results demonstrate that our proposed adaptation mechanism better unleashes the potential of diffusion models in few-shot dense tasks.

## 2 RELATED WORK

**Few-shot learning.** Few-shot learning traditionally focuses on learning new categories in tasks such as classification, detection, and segmentation. Siamese networks aim to improve matching by learning better embeddings, while MAML treats few-shot adaptation as an optimization objective (Finn et al., 2017). Subsequently, evidence suggests that fine-tuning well-pretrained representations yields better results than meta-learning methods (Tian et al., 2020). Our work investigates how to fine-tune pretrained diffusion representations for downstream dense prediction tasks. However, prior research on few-shot dense prediction has primarily targeted segmentation and focused on learning new categories. To the best of our knowledge, VTM is the only work addressing few-shot learning for universal dense prediction. However, it relies on meta-learning, which requires significant dense annotations for multiple tasks.

**Parameter-efficient fine-tuning.** Fine-tuning originally involved optimizing the entire set of model parameters, even with limited samples. However, as model sizes grow, optimizing all parameters

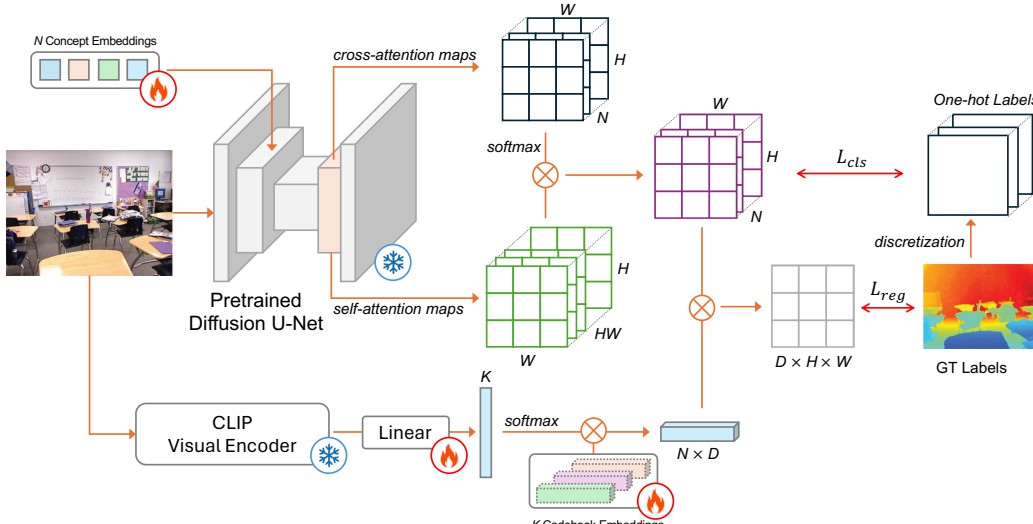

Figure 1: We reformulate all dense prediction tasks into an $N$-class classification task. Specifically, we optimize $N$ concept embeddings, enabling the pre-trained diffusion model to produce a probability distribution across $N$ classes for each pixel. This is achieved by combining the model's internal attention masks. For tasks with continuous outputs, we use a learnable label codebook to perform label-to-value mapping, making the mapping input-dependent to handle one-to-many uncertainties. Our method optimizes only a few parameters, leveraging both classification and regression losses. This design facilitates few-shot learning for dense prediction tasks.

often results in severe overfitting. Consequently, there is a shift toward designing parameter-efficient fine-tuning (PEFT) methods. Popular techniques include LoRA (Hu et al., 2021), widely used in large language models (LLMs) and stable diffusion optimization. However, LoRA-like methods still optimize parameters across many layers. To address this, strategies like VPT (Jia et al., 2022) and Bias Tuning (Cai et al., 2020) focus on optimizing only input embeddings or bias parameters. These approaches are more efficient, offer better interpretability, and mitigate overfitting. Our method aligns with this paradigm by solely optimizing concept embeddings, replacing the text inputs of pretrained diffusion models.

**Generalist models.** Recent advancements in computer vision have introduced "generalist models," capable of handling multiple tasks (Wang et al., 2023a;b; Kirillov et al., 2023). Unlike traditional multi-task learning, which typically shares only encoder parameters, generalist models share all parameters across tasks. During training, they jointly learn from multiple tasks, and at inference, they adapt to different tasks by altering the input prompt without modifying model parameters. Prompts can take various forms, such as geometric inputs (e.g., points or boxes), text, or pairs of images and targets. However, these prompt-based models are limited to handling tasks encountered during training. Even when provided with a few annotated examples as prompts, adapting to entirely unseen tasks remains challenging. At best, they can quickly adjust to new classes within known tasks, akin to how a painter adapts to variations within a familiar style. In contrast, we argue that a truly foundational vision model should adapt easily to a wide range of unseen tasks, leveraging extensive pretraining. This distinction underpins the primary difference between our approach and generalist models.

**Generative models for perception.** There is a growing trend to use generative models for inherently discriminative tasks. Examples include Dataset GAN (Zhang et al., 2021), SemanticGAN (Li et al., 2021), ODISE (Xu et al., 2023), and VPD (Zhao et al., 2023), among others (Bhattad et al., 2024; Du et al., 2023). These methods enhance discriminative tasks by generating images, modifying the generative model for novel tasks, or introducing specialized layers and decoders for specific scene properties. Similarly, we leverage a pretrained diffusion model for dense perception tasks. Unlike prior methods, our approach introduces a unified architecture for multiple dense tasks and demonstrates the ability to adapt to new tasks with only a few examples.

# 3 METHOD

Our method demonstrates that diffusion models can be adapted for dense vision tasks with a few training examples. Before we detail our method, we first formulate the few-shot adaptation problem and revisit latent diffusion objective and architecture.

**Problem formulation.** Given an input image $\mathcal{I}$, a pretrained latent diffusion model $\mathcal{M}$, and a dense prediction task $\mathcal{T}$, our goal is to generate an output $\mathcal{J}$. $\mathcal{J}$ has the same height $H$ and width $W$ dimensions as the original image $\mathcal{I}$ but its values can be continuous or discrete. If $\mathcal{T}$ is a semantic segmentation task across $K$ categories, the output is a binary tensor $\mathcal{J} \in \mathbb{R}^{H \times W \times K}$. If $\mathcal{T}$ is a depth estimation task, the output is a continuous tensor $\mathcal{J} \in \mathbb{R}^{H \times W}$.

**Background on diffusion.** Text-to-image diffusion models are trained to produce photorealistic images conditioned on text inputs. They are trained using large scale image-text datasets scraped from the internet (Schuhmann et al., 2022) They generate images from random Gaussian noise through a sequence of $T$ denoising steps (Ho et al., 2020). The denoising process is defined as a sequence of timesteps $T, T-1, ..., 0$. At timestep $T$, a $H \times W \times 3$-dimensional random noise is sampled from a multivariate normal distribution, denoted as the initial noise $\boldsymbol{x}_T$. DDPM applies a denoising neural network to iteratively de-noise latents $x_{t+1} \rightarrow \boldsymbol{x}_t$ until $\boldsymbol{x}_0$.

Denoising in pixel space is computationally expensive as each denoising step produces a $H \times W \times 3$-dimensional noise estimation. **Latent Diffusion Models** (Rombach et al., 2022), $\mathcal{M}$, instead encode the sample image $\boldsymbol{x}$ into a much smaller latent space $\boldsymbol{z}_0$ using a pretrained variational autoencoder (VAE) (Esser et al., 2020) with a $h \times w \times c$-dimensional hidden representation, such that $h \times w \times c \ll H \times W \times 3$. Latent diffusion models sample $\boldsymbol{z}_T$ from a random Gaussian distribution and iteratively denoise in the latent space until $\boldsymbol{z}_0$. Finally, $\boldsymbol{z}_0$ is decoded using the pretrained VAE's decoder into the pixel space.

## 3.1 REFRAMING DENSE VISION TASKS AS CLASSIFICATION

We reformulate all dense prediction tasks into classification problems. Some dense prediction tasks, such as segmentation, inherently have outputs that are suitable for discrete classification. However, tasks like depth estimation, whose outputs are continuous values, cannot be directly optimized through classification loss. We discretize a continuous output space of size $H \times W \times D$. First, we split the possible range of continuous values in the original $D$-dimensional output space into $N$ buckets. We assign a unique ID from $\{0, \ldots, N-1\}$ to each bucket. Each ground truth continuous value is discretized into the bucket with its corresponding range; the ID of this bucket is used as the category label for the classification reformulation.

Intuitively, for depth estimation, we divide the physical measurement of the continuous depth values into $B$ buckets. Semantically, categories represent concepts like "very close" or "a little close", "very far", etc. We adapt Stable Diffusion with a few training examples to understand these new categories.

## 3.2 REFRAMING DENSE VISION TASKS AS CLASSIFICATION

We reformulate all dense prediction tasks into classification problems. Some dense prediction tasks, such as segmentation, inherently have outputs that are suitable for discrete classification. We label their ground truth as $GT_{\text{label}}$. However, tasks like depth estimation, whose outputs are continuous values, cannot be directly optimized through classification loss. For such tasks, we label their original ground truth as $GT_{\text{value}}$. For these continuous-output tasks, we discretize their ground truth $GT_{\text{value}}$ of size $H \times W \times D$ into $GT_{\text{label}}$. This gives them both $GT_{\text{value}}$ and $GT_{\text{label}}$ as ground truth targets, enabling the use of both regression loss and classification loss during training.

Specifically, we split the possible range of continuous values in the original $D$-dimensional output space into $N$ buckets. Then, we assign a unique ID from $\{0, \ldots, N-1\}$ to each bucket. Each ground truth continuous value is discretized into the bucket corresponding to its range, and the ID of this bucket is used as the ground truth category label for the classification reformulation.

Intuitively, for depth estimation, we divide the physical measurement of the continuous depth values into $N$ buckets. Semantically, these categories represent concepts like "very close," "a little close," or "very far." We adapt Stable Diffusion with a few training examples to understand these new categories.

## 3.3 Concept embeddings for extracting Stable Diffusion representations

Although diffusion models are trained with the objective of denoising photorealistic images, their learned internal representations encode sufficient semantic and geometry information. To extract meaningful representations, we learn $N$ task-specific concept tokens $\{T_1, \ldots, T_N\}$ to replace stable diffusion's text tokens. For an input image $I$ and task $\mathcal{T}$, we load its corresponding concept embeddings as inputs to $\mathcal{M}$. Our goal is to design a model that generates a probability distribution over each pixel of the input image $I$ across the $N$ categories. Therefore, we need to output a $H \times W \times N$ output.

Stable diffusion designs $\mathcal{M}$ as a UNet. The UNet architecture employs two types of attention modules: self-attention and cross-attention. An attention module contains three components: query, key, and value. In the self-attention module, all three components are derived from the image latents; each latent variable attends over all the latent variables, capturing the image's global structure. The self-attention masks for a layer with height $H'$ and width $W'$ are $M_{sa} \in \mathbb{R}^{H' \times W' \times (H' \times W')}$ In the cross-attention module, the cross-attention module calculates cross-modal associations between the image latents and the input text. This process allows the model to ensure that the generations are related to the text conditioning. Therefore, the cross-attention masks are $M_{ca} \in \mathbb{R}^{H' \times W' \times T}$, where $T$ is the number of text tokens.

Conditioned on the concept embeddings and an input image, $\mathcal{M}$ produces $\{M_{sa}^1, \ldots, M_{sa}^n\}$ and $\{M_{ca}^1, \ldots, M_{ca}^n\}$ where $n$ is the number of layers. Attention masks at different layers allow us to access information at various points in the feature hierarchy. $M_{ca}$ includes $N$ masks of size $H' \times W'$, each indicating the relationship between newly-defined classes across all the positions on the image. $M_{sa}$ includes $H'W'$ masks of size $H' \times W'$, each indicating the relationship with all other image latents. By combining these two, we can capture the relationship between image latents and between image latents and the concept embeddings. We first normalize $M_{ca} \in \mathbb{R}^{(H'W') \times N}$ and then use it as the weight to aggregate $M_{sa} \in \mathbb{R}^{(H'W') \times H' \times W'}$. From this, we get an attention mask $M_{attn} \in \mathbb{R}^{H' \times W' \times N}$ through $M_{attn} = M_{sa}^T M_{ca}$.

We can obtain a different $M_{attn}$ from every layers in the UNet. We further combine them by first upsampling each attention mask to $H \times W \times N$. Then, we simply average all of them across layers and subsequently normalize them across $N$. Finally we get $M_{label} \in \mathbb{R}^{H \times W \times N}$, which is the probability distribution across $N$ task-specific categorizes for each pixel of $I$.

## 3.4 Learnable codebook-conditioned classification

Although we formulate continuous dense tasks as classification problems, their final output still needs to be continuous. Therefore, for these tasks, we need to project the category distribution $M_{label}$ to the continuous output $M_{value} \in \mathbb{R}^{H \times W \times D}$. However, mapping from discrete class labels to continuous values in such tasks is an ill-posed one-to-many transformation compared to the mapping from continuous values to categories.

To address this, we represent the label-to-value mapping as a learnable random variable. The range of these random variables needs to be conditioned on the input image. For instance, when representing depth, the values should be smaller for indoor scenes but larger for outdoor images. We model this random variable as a learnable codebook $\mathcal{C}$, which contains $K$ sets of mappings from labels to values. Each label-to-value mapping of size $N \times D$ represents how $N$ discrete labels correspond to $D$-dimensional continuous output values. Thus, $\mathcal{C} \in \mathbb{R}^{K \times N \times D}$.

For an input image $I$, we first calculate its distribution over the $K$ associated mappings. Specifically, we use a frozen pre-trained CLIP visual encoder to extract a $D'$-dimensional feature from the input image. Then, through a learnable linear mapping $\mathcal{L}$ of size $D' \times K$, this extracted feature is mapped to a $K$-dimensional vector. After applying softmax, we obtain a distribution $F$ of the mapping relationship conditioned on the input. Using $F$ as weights, we combine the $K$ sets of mapping relationships $\mathcal{C}$ through a weighted sum to obtain $\mathcal{C}' \in \mathbb{R}^{N \times D}$. This represents how the $N$ discrete labels map to continuous output values for the input image.

Finally, based on the obtained $\mathcal{C}'$, the sample-dependent label-to-value mapping, we use the previously obtained per-pixel $N$-class probability distribution $M_{label}$ to calculate the expected value at each pixel position. This process converts the probability distribution over discrete labels into continuous

values. This produces $M_{value}$, which lies in the original output space for tasks with continuous values, such as depth estimation.

## 3.5 TRAINING AND INFERENCE

Using the obtained $M_{label}$ and $M_{value}$, as well as the ground truth targets $GT_{label}$ and $GT_{value}$, we train the model by optimizing both the classification and regression losses. The whole optimization objective can written as:

$$L_{\text{cls}} = \text{CrossEntropy}(M_{label}, GT_{label}) \qquad L_{\text{reg}} = \text{L2}(M_{value}, GT_{value}) \qquad (1)$$
$$L = L_{\text{cls}} + \alpha L_{\text{reg}} \qquad (2)$$

To avoid overfitting with few-shot examples, our method is designed to include only a limited number of parameters: the concept embeddings $\{T_1, \ldots, T_N\}$, the label codebook $\mathcal{C}$, and the linear mapping $\mathcal{L}$. For tasks with discrete outputs, such as segmentation, we directly output $M_{label}$ without applying the learnable codebook. In this case, we train the model only with $L_{\text{cls}}$.

## 4 EXPERIMENTS

In this section, we first describe our experimental setup and evaluation protocol. Then, we compare our model's ability to adapt to unseen dense vision tasks using a few examples with previous methods. Finally, we discuss our design choices and comprehensively study the characteristics of our model.

### 4.1 SETUP

**Data.** To effectively evaluate our model, we consider two datasets: Taskonomy (Zamir et al., 2018) and NYUv2 (Silberman et al., 2012). Taskonomy (Zamir et al., 2018) comprises 10 dense vision tasks, including Euclidean distance, Z-buffer depth, texture edge, occlusion edge, 2D keypoints, 3D keypoints, reshading, principal curvature, surface normal, and semantic segmentation. It allows us to validate the model's capability of adapting to a diverse range of tasks, each requiring a different form of understanding. To further study the model's geometric understanding capability, we additionally include two tasks from the NYUv2 dataset (Silberman et al., 2012): depth estimation and surface normal prediction.

**Metrics.** We employ different metrics for each task. Following previous work (Kim et al., 2023), we utilize root mean square error (RMSE) to evaluate Euclidean distance, Z-buffer depth, texture edge, occlusion edge, 2D keypoints, 3D keypoints, reshading, principal curvature, and NYU depth. We adopt mean intersection over union (mIoU) for semantic segmentation. For surface normal, we use mean error (mErr).

**Evaluation protocol.** Few-shot adaptation outcomes can vary widely depending on the training examples used. To address this variability, we conduct each few-shot experiment 100 times with different randomly selected training data points and compute the average. We note that this setup, while necessary, is absent in previous work (Kim et al., 2023).

**Baselines.** We compare our model against VTM (Kim et al., 2023) and VPD (Zhao et al., 2023). VTM (Kim et al., 2023) is a state-of-the-art few-shot dense predictor, while VPD (Zhao et al., 2023) is a few-shot model that also utilizes diffusion priors. To understand the potential upper limits of these methods, we also conduct experiments under fully supervised conditions. For more details, please refer to the supplementary material.

**Implementation details.** We use Stable Diffusion 2.1 as the pre-trained backbone to extract features for both our method and the VPD baseline. We extract cross-attention and self-attention attention masks from the 8th to the 12th layers, and the last three layers of the UNet. We adopt different numbers of discretization buckets for different tasks. Specifically, we set the bucket size to 20 for texture edge, occlusion edge, 2D keypoints, 3D keypoints, and reshading. We adopt 40 buckets for principal curvature, Euclidean distance, Z-buffer depth, and surface normal estimation. For NYUv2, we use 50 buckets.

For tasks with 1-dimensional outputs, we transform their value range into a uniform log space. For tasks with multidimensional outputs, we follow previous work (Wang et al., 2015) to partition the output space using k-means and Delaunay triangulation. The values obtained from these divisions were used as the initial values for each label distribution in the learnable label codebook. We randomly initialize the query embedding. During optimization, we randomly select a timestep between 5 and 200, and add noise of corresponding intensity to the latent input to the UNet, based on the noise scheduler of stable diffusion. During the inference phase, we consistently set this timestep to 200. For more optimization details, We leave them to the supplementary material.

Table 1: We compare the performance of the diffusion prior with DINOv2 and CLIP on the NYUv2 depth estimation and surface normal estimation benchmarks under the few-shot setting. The results show that the diffusion prior has a clear advantage over contrastive-based pre-trained models in few-shot adaptation for dense tasks. Additionally, we observe that the learnable label-to-value mapping (Diffusion Learnable) outperforms the fixed label-to-value mapping (Diffusion Fixed).

| Method | CLIP | DINOv2 | Diffusion Fixed | Diffusion Learnable |
|---|---|---|---|---|
| Depth Estimation↓ | 0.64 | 0.57 | 0.51 | 0.47 |
| Surface Normal Estimation↓ | 23.4 | 18.3 | 17.8 | 17.2 |

## 4.2 DIFFUSION PRIOR AS A STRONG BASELINE FOR FEW-SHOT DENSE TASKS

We first demonstrate that the pre-trained diffusion model alone already serves as a strong baseline for few-shot dense tasks. To isolate the impact of the diffusion model from that of the CLIP model, we set the codebook size to 0 or 1. A codebook size of 0 represents a pre-defined fixed label-to-value mapping, while a size of 1 represents a learnable label-to-value mapping without requiring the CLIP encoder to determine how to combine different label-to-value results.

In addition, we compare our approach to CLIP and DINOv2. Similar to diffusion models, both were pre-trained on large-scale datasets. However, unlike our model, they were trained using contrastive objectives. We use a pre-trained ViT-L (Dosovitskiy et al., 2020) as the encoder and attach a trainable linear projection layer. We transform the continuous outputs of depth and surface normals into discrete classification problems as well. For ease of comparison, we used a fixed pre-defined class-to-value mapping instead of a learnable label codebook.

We compare the impact of using different pre-trained models as backbones on the NYU depth estimation and surface normal prediction tasks, with only 20 examples provided. We use RMSE to evaluate depth estimation and mErr to evaluate surface normal estimation.

As shown in Table. 1, the CLIP and DINOv2 backbones perform worse than the diffusion backbone under the few-shot dense prediction setting. We conjecture that this may be partly because the pre-trained diffusion model, being generatively pre-trained, retains more detailed information compared to contrastive loss-based pre-training, making it more suitable for few-shot adaptation. Furthermore, we observe that the learnable class-to-value mapping outperforms the fixed pre-defined class-to-value mapping.

## 4.3 RESULTS ON 12 DENSE TASKS

**Quantitative results.** Building on the already strong baseline of the diffusion prior for few-shot dense tasks, we further incorporated the codebook design to enhance performance on these tasks. We conducted experiments using the full model on 10 Taskonomy tasks and 2 NYU tasks. For the Taskonomy tasks, we followed  Kim et al. (2023) and used a 10-shot few-shot setting. For the NYUv2 tasks, we used a 20-shot few-shot setting.

As shown in Table 2, compared to methods specifically designed for few-shot segmentation, multi-task learning-based methods, and methods that use the diffusion prior without special design for few-shot settings, our model significantly outperforms them on NYUv2, and achieves slightly better performance on Taskonomy. We, however, note that VTM (Kim et al., 2023) divides the 10

Table 2: Few-shot results on 10 tasks from Taskonomy and 2 tasks from NYUv2. Lower values indicate better performance across all tasks except for semantic segmentation (SemSeg). For VTM, VPD, and our method, we report the average values after running 100 iterations. For HSNet, VAT, and DGPNet, we directly copied the results from Kim et al. (2023).

| Methods | EucDepth↓ | Z-depth↓ | 2DEdge↓ | 3DEdge↓ | 2DKeypoint↓ | 3DKeypoint↓ |
|---|---|---|---|---|---|---|
| HSNet Min et al. (2021) | 0.2375 | 0.0748 | 0.1746 | 0.1643 | 0.1056 | 0.0651 |
| VAT Hong et al. (2022) | 0.2718 | 0.0779 | 0.1719 | 0.1655 | 0.1450 | 0.0678 |
| DGPNet Johnander et al. (2022) | 0.4579 | 0.2846 | 0.1881 | 0.2130 | 0.1104 | 0.1308 |
| VTM Kim et al. (2023) | 0.0812 | 0.0347 | 0.0818 | 0.0917 | 0.0671 | 0.0512 |
| VPD Zhao et al. (2023) | 0.1056 | 0.0404 | 0.0965 | 0.1226 | 0.0697 | 0.0670 |
| Ours | **0.0776** | **0.0308** | **0.0625** | **0.0812** | **0.0626** | **0.0389** |

| Methods | Reshading↓ | Curvature↓ | Normal↓ | SemSeg↑ | NYUDepth↓ | NYUNormal↓ |
|---|---|---|---|---|---|---|
| HSNet Min et al. (2021) | 0.2627 | 0.0610 | 24.9120 | 0.1069 | – | – |
| VAT Hong et al. (2022) | 0.2709 | 0.0796 | 25.8134 | 0.353 | – | – |
| DGPNet Johnander et al. (2022) | 0.3680 | 0.3574 | 29.1668 | 0.0261 | – | – |
| VTM Kim et al. (2023) | 0.1308 | 0.0413 | 11.7850 | 0.3980 | 0.73 | 26.1 |
| VPD Zhao et al. (2023) | 0.1609 | 0.0498 | 14.4381 | 0.3484 | 0.49 | 18.5 |
| Ours | **0.1284** | **0.0376** | **10.1346** | **0.4178** | **0.43** | **16.4** |

Taskonomy tasks into 5 groups, and pre-train their model on four of these groups using multitask learning, before few-shot adaption on the remain two tasks. Therefore, VTM effectively trains five separate models on Taskonomy, and the model used for testing varies across different tasks. In contrast, we apply the same model for all few-shot adaption tasks, and our model does not require any additional pre-training. Our model also compares favorably against VPD (Zhao et al., 2023), as the large number of parameters in VPD hinder its ability to adapt to new tasks quickly.

**Qualitative results.** We visualize the dense prediction results in Figure 2. Compared to prior art, our model captures more fine-grained details, such as the lights on the ceiling and the legs of the chairs in the background.

## 4.4 ANALYSIS

Table 3: We compare the influence of using self attention mask $M_{sa}$ on NYU depth estimation and surface normal prediction with 20 examples are given. We report RMSE for depth estimation and mErr for surface normal estimation. The results indicate combining self-attention and cross-attention masks as the mask proposal from the diffusion backbone provides a better representation of the input image compared to using only the cross-attention mask.

|  | w/o $M_{sa}$ | w/ $M_{sa}$ |
|---|---|---|
| Depth Estimation↓ | 0.47 | **0.43** |
| Surface Normal Estimation↓ | 17.4 | **16.4** |

**Do we need the self-attention masks?** To verify the necessity of utilizing self-attention masks for mask proposals, we conduct an ablation (see Table 3). Results suggest that including self-attention effectively reduces the RMSE for various tasks.

**Is classification really needed?** We further investigate the necessity of transforming regression tasks into classification tasks. Specifically, we optimized only one text embedding, producing an output of

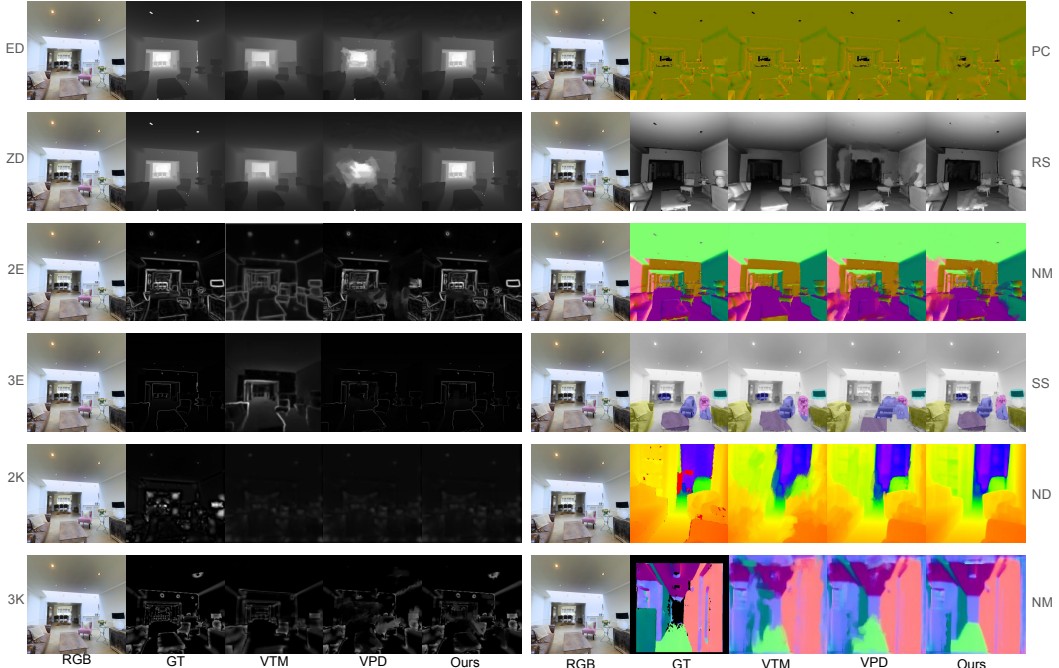

Figure 2: We show the visualization results of our method on different tasks under the few-shot setting. Our method often accurately captures finer-grained details, such as the lights on the ceiling, which are overlooked by other few-shot dense prediction methods.

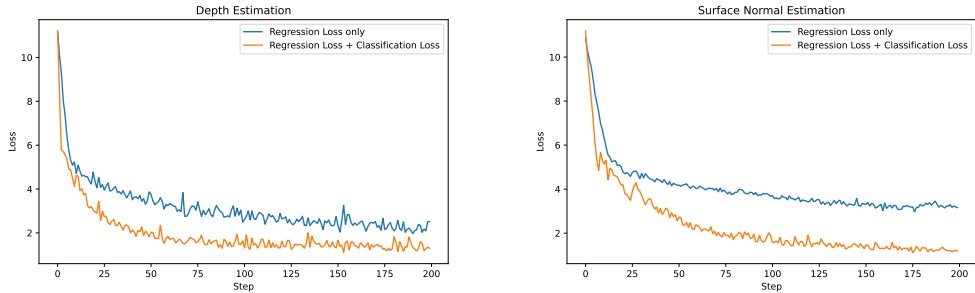

Figure 3: We demonstrate the necessity of classification loss by conducting a fast few-shot adaptation experiment with only 20 examples on NYU depth and normal tasks. We compare the impact of classification loss on the optimization process and use a fixed label-to-value mapping to calculate the RMSE metric on the validation set. The results show that classification loss facilitates faster convergence under this few-shot setting.

size $H \times W$. We then normalized the output value at each pixel's location to the range of $[0, 1]$ and compute the regression loss with the ground-truth real continuous value on validation set. As shown in Figure 3, incorporating the classification loss results in faster and better convergence.

**How large the learnable value codebook should be?** The last question we address is whether a learnable value codebook is needed to map from discrete categories to continuous real value outputs. To this end, we compare the impact of using a predefined fixed mapping versus different codebook sizes of 1, 5, 10, 25, and 50 on the performance of NYUv2 depth estimation and surface normal prediction. The results are reported in Table 4. We find that learnable mapping performs better than fixed mapping. The codebook does not need to be very large; constantly increasing the codebook size does not lead to consistent improvements and may instead degrade performance. We observed that a codebook size of around 10 tends to offer near-optimal performance. Therefore, we set the codebook size to 10 for all tasks.

Table 4: We compare the different sizes of learnable label codebook on NYU depth estimation and surface normal prediction with 20 examples are given. We reporte RMSE for depth estimation and mErr for surface normal estimation. N/A refers to the pre-defined fixed label-to-value mapping, where the size of learnable code book is 0. We can see that both excessively large and excessively small codebook sizes lead to a decline in performance.

|  | N/A | 1 | 5 | 10 | 25 | 50 |
|---|---|---|---|---|---|---|
| Depth Estimation↓ | 0.51 | 0.47 | 0.44 | **0.43** | 0.45 | 0.48 |
| Surface Normal Estimation↓ | 17.8 | 17.2 | 16.9 | **16.4** | 17.4 | 17.6 |

## 5  DISCUSSION

**Limitations.** Although we demonstrate that the diffusion prior can quickly adapt to new tasks by training with few-shot examples, we identify several limitations of the current methods. First, when converting continuous output tasks into classification tasks, we divide the output into varying numbers of categories depending on the task. Consequently, we learn different numbers of concept embeddings for different tasks. Furthermore, for tasks with different output dimensions, such as depth and normal prediction, we adopt distinct methods to partition the output space. However, these strategies essentially rely on human prior knowledge about the tasks to facilitate fast adaptation. Ideally, we aim to rely solely on the model's inherent priors, without depending on human task-specific knowledge. This would enable the model to quickly adapt to new tasks using only a few examples.

**Future Direction.** Our goal is to enable visual models to learn new tasks quickly from a small number of samples, akin to human learning. Ideally, this would be achieved through in-context learning, where the model grasps the definition of unseen tasks without requiring task-specific fine-tuning, relying solely on a few annotated examples. While LLMs achieve this for many language tasks, few-shot in-context learning for unseen dense vision tasks remains challenging due to the diversity in output formats across dense tasks, as well as the vastly different knowledge requirements for tasks like semantic segmentation (semantic understanding) and depth estimation (geometric reasoning).

This work takes a step toward this goal by relaxing the no-fine-tuning restriction, exploring how diffusion models can adapt to new tasks using only a few examples with minimal parameter changes. We focus on modifying input parameters rather than internal ones, aligning with in-context adaptation. Unlike task-specific heads, this approach enables task differences to be defined through inputs, which is closer to in-context learning. Our findings show that diffusion models can adapt to unseen dense tasks by optimizing only input parameters, demonstrating their potential as general decoders for task definitions via inputs.

Future directions include learning task definitions from few-shot samples through feedforward mechanisms, such as using autoregressive models to learn concept embeddings (task vectors) that prompt the diffusion decoder. Another promising direction is studying relationships between tasks represented by these task vectors. We aim to develop a promptable interface for downstream task adaptation without gradient-based backpropagation, leveraging the diffusion prior as a foundation.

**Conclusion.** In this work, we repurpose stable diffusion into a backbone that can adapt to various dense prediction tasks with only a few training samples. The key lies in transforming tasks with different output spaces into classification tasks, as this simplifies the optimization process. By utilizing the different levels of attention masks within stable diffusion, we have transformed the pre-trained generation model into a mask proposal generator. Furthermore, we have learned a codebook to map the category labels obtained from classification back to the original numerical output space. Extensive experiments have demonstrated that our method successfully adapts the priors learned by the pre-trained diffusion model to various perception tasks through few-shot learning.

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

# APPENDIX

## A  BROADER IMPACT

Our method, which employs diffusion for general few-shot dense tasks, offers significant advantages beyond technical improvements. It substantially reduces labor costs associated with pixel-by-pixel annotation of visual dense tasks, making model deployment more cost-effective and accessible, especially for resource-limited projects. Additionally, the few-shot nature of our approach reduces energy consumption, lowering the environmental impact by decreasing the need for extensive data and computational resources. This aligns with broader goals of energy conservation and emission reduction. By democratizing access to advanced machine learning technologies, our method enables smaller entities and individuals to innovate and implement AI solutions, promoting more responsible and ethical AI development.

## B  RESULTS ON FULLY TRAINING SET

We include the results of full training set in Table. 5. Although VPD's performance in the few-shot setting is not strong, with more training data, we can see that its performance improves significantly because it fine-tunes more parameters. In contrast, we only fine-tune the concept embeddings with a few hundred parameters. However, our method still outperforms VTM even after training on the full training set, demonstrating the higher potential of the diffusion prior. We also reported the 95% confidence interval, and it can be seen that our method, leveraging a very general prior, achieved more stable results compared to VTM.

## C  RESULTS WITH DIFFERENT NUMBER OF TRAINING SAMPLES

In Fig 4, we illustrate the impact of using 10, 20, 50, and 100 training samples on our method and VPD across all 12 tasks. It can be observed that our method consistently adapts better to new tasks compared to VPD when fewer than 100 training examples are provided. Moreover, as the number of training samples increases, the performance of both methods improves accordingly.

Table 5: We present the results on 10 tasks from Taskonomy and 2 tasks from NYUv2. For Taskonomy tasks, 10-shot training examples are used for each of them, and for NYU tasks, we use 20 examples. To also evaluate the statistical robustness, we run each number for 100 times and report the 95% confidence interval. Besides segmentation task, lower number indicates better performance. Our method consistently outperforms VTM on all few-shot tasks, especially on out-of-domain tasks. And our method better unleashes the power of diffusion prior for few-shot dense prediction compared to VPD.

| | Few-shot | | | Fully Supervised | | |
|---|---|---|---|---|---|---|
| | VTM | VPD | Ours | VTM | VPD | Ours |
| EucDepth | 0.0812 ±0.0065 | 0.1056 ±0.0102 | **0.0776** ±0.0072 | 0.0524 | 0.0456 | 0.0498 |
| Z-depth | 0.0347 ±0.0035 | 0.0404 ±0.0037 | **0.0308** ±0.0038 | 0.0257 | 0.0210 | 0.0236 |
| 2DEdge | 0.0818 ±0.0021 | 0.0965 ±0.0023 | **0.0625** ±0.0022 | 0.0154 | 0.0131 | 0.0136 |
| 3DEdge | 0.0917 ±0.0028 | 0.1226 ±0.0044 | **0.0812** ±0.0040 | 0.0638 | 0.0564 | 0.0599 |
| 2DKeypoint | 0.0671 ±0.0038 | 0.0697 ±0.0035 | **0.0626** ±0.0040 | 0.0337 | 0.289 | 0.306 |
| 3DKeypoint | 0.0512 ±0.0018 | 0.0670 ±0.0027 | **0.0389** ±0.0014 | 0.0360 | 0.0298 | 0.0324 |
| Reshading | 0.1308 ±0.0058 | 0.1609 ±0.0044 | **0.1284** ±0.0049 | 0.834 | 0.756 | 0.772 |
| Curvature | 0.0413 ±0.0010 | 0.0498 ±0.0019 | **0.0376** ±0.0023 | 0.0345 | 0.0291 | 0.329 |
| Normal | 11.7850 ±0.4580 | 14.4381 ±0.3097 | **10.1346** ±0.0361 | 6.2418 | 5.7963 | 5.9821 |
| SemSeg | 0.3980 ±0.0350 | 0.3484 ±0.0308 | **0.4178** ±0.0361 | 0.4618 | 0.4905 | 0.4784 |
| NYUDepth | 0.73 ±0.09 | 0.49 ±0.11 | **0.43** ±0.08 | 0.35 | 0.25 | 0.29 |
| NYUNormal | 26.1 ±3.8 | 18.5 ±1.7 | **16.4** ±1.6 | 18.2 | 14.8 | 14.9 |

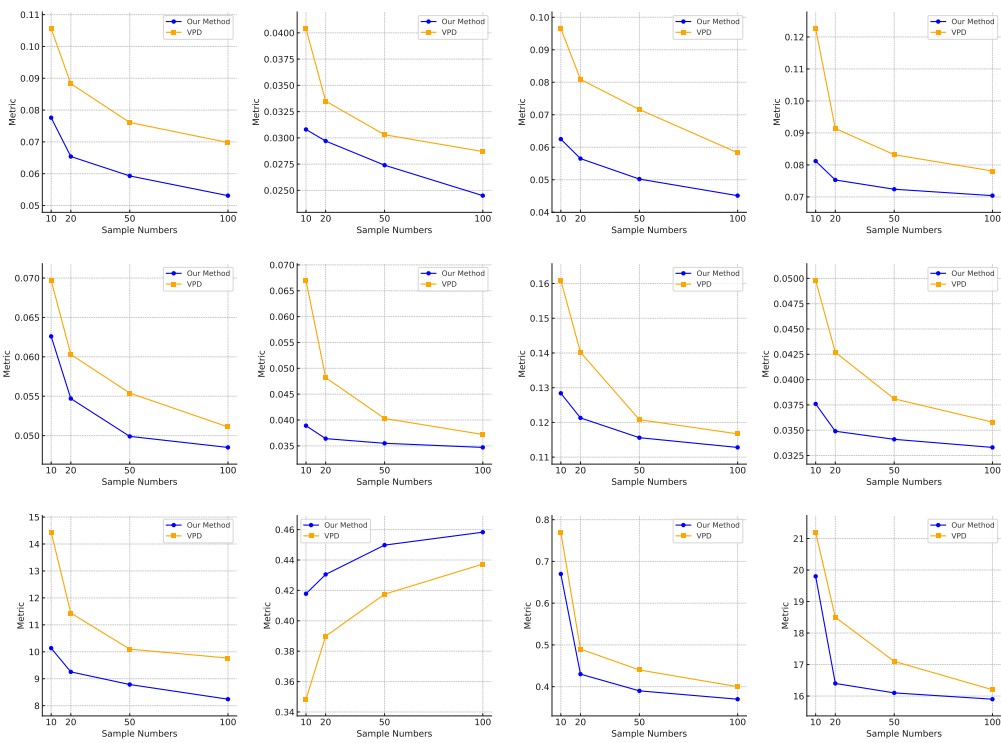

Figure 4: We present the impact of using different numbers of training samples on our method and VPD across all 12 tasks.