# OpenReview forum: "Diffusion Models are Few-shot Learners for Dense Vision Tasks"
_ICLR.cc/2025/Conference — Submitted to ICLR 2025_

### Official Review · Reviewer_u6mW · 2024-10-28

**Soundness:** 3
**Presentation:** 3
**Contribution:** 3
**Rating:** 6
**Confidence:** 4

**Summary:**

The paper proposes a general strategy to adapt pre-trained large diffusion models to dense image tasks (both classification and regression) in a few-shot fashion. The tackle this by framing all dense problems as pixel-wise classification and learning a a set of conditioning encodings for the cross attention, that are responsible for the classification. To address the regression tasks, the authors introduce a codebook that serves as a basis to construct the real value output, and pick from that codebook in an image-adaptive fashion.
The results show SoTA performance on Taskonomy and NYUv2 datasets.

**Strengths:**

- The proposed technique is simple and allows to effectively bring out the knowledge out of a large diffusion model for the classification task.
- SoTA on two standard benchmark datasets.
- The ablation experiments clearly demonstrate the importance of every component.

**Weaknesses:**

- It's not clear what the rows in Figure 2 mean.
- See some other concerns in questions.

**Questions:**

- Why do you perform inference of a diffusion U-Net with noise at t=200? I would expect the noise to degrade the performance for dense prediction tasks. Why not use a lower level of noise, e.g., t=1? Also during training time t is randomly sampled in [5, 200] range, why not make it consistent with the inference?
- It seems like 10 or 20 examples for training the codebook, the queries and the linear projection layer for clip is a little small. Have you observed overfitting? If so how have you tried addressing it?
- Does figure 3 show training loss? If so, it would be great to look at the validation loss instead.

---

> ### Author Response · Authors · 2024-11-29
>
> We appreciate the reviewers for their valuable feedback.
>
> ---
>
> **Q1: Noise added to the input**
>
> We found that adding noise helps extract better representations from the pre-trained diffusion model, as demonstrated in [1]. In [1], features extracted at \(t=234\) encode better visual correspondences compared to those at lower noise levels (e.g., \(t=1\)). During training, we randomly sampled \(t\) in the \([5, 200]\) range as an augmentation strategy to prevent overfitting.
>
> ---
>
> **Q2: Risk of overfitting**
>
> Although we only used 10 or 20 training examples, we did not observe significant overfitting. This is partly because we applied several augmentation strategies, such as adding noise at different levels. Additionally, the number of parameters we fine-tuned is relatively small. Furthermore, each training example includes pixel-level annotations, where every pixel has a corresponding ground truth label. This means the number of annotations is much larger compared to datasets where each image has a single class label.
>
> However, we did notice overfitting when the codebook size was too large, so we carefully controlled the codebook size. There is still some degree of overfitting, as increasing the number of training samples tends to improve validation performance even when the training loss remains similar. Nonetheless, the overfitting is not severe, especially compared to other methods.
>
> ---
>
> **Q3: Clarification regarding Figure 3**
>
> The vertical axis of Figure 3 represents the evaluation metric computed on the validation set. We compared the impact of using classification loss during training on validation metrics. Since the evaluation metric for continuous dense tasks (RMSE) coincidentally matches the regression loss, referring to it as "regression loss on the validation set" caused ambiguity. This has been clarified in the revised version.
>
> ---
>
> **W1: Explanation of the rows in Figure 2**
>
> Each row in Figure 2 contains two tasks. For each task, there are five images from left to right: the input, ground truth, VTM prediction, VPD prediction, and our method's prediction.
>
> ---
>
> [1]. Emergent Correspondence from Image Diffusion
> Luming Tang*, Menglin Jia*, Qianqian Wang*, Cheng Perng Phoo, Bharath Hariharan

---

> > ### Comment · Reviewer_u6mW · 2024-11-29
> >
> > I thank the authors for the clarifications. I keep my score of 6.

---

> > > ### Author Response · Authors · 2024-12-02
> > >
> > > Thank you for your thoughtful comments and for taking the time to review our work. We truly appreciate your valuable feedback and your acknowledgment of our clarifications.

---

### Official Review · Reviewer_66Ct · 2024-10-30

**Soundness:** 2
**Presentation:** 1
**Contribution:** 2
**Rating:** 3
**Confidence:** 5

**Summary:**

This paper presents an innovative approach to adapt pretrained diffusion models for few-shot learning in dense vision tasks, such as depth estimation and semantic segmentation. By reframing dense prediction tasks as codebook-conditioned classification problems, the authors enable the model to handle various tasks without altering internal parameters. They introduce concept and codebook embeddings to enhance task-specific adaptation, achieving notable performance on 12 vision tasks, especially in low-data scenarios.

**Strengths:**

1. The use of pretrained diffusion models for few-shot dense tasks is sound, leveraging generative models for discriminative tasks without modifying model parameters.
2. The model shows state-of-the-art results across multiple benchmarks, outperforming other few-shot learning methods like VTM and VPD.
3. The approach reduces training data requirements, which could lead to lower computational costs and broader accessibility.

**Weaknesses:**

1. The overall presentation of this paper is poor, making it hard to understand. Specifically, the notations are not clearly defined and some of them may cause confusion, for example, n, N, K.

2. The operations and the rationale behind it are not clearly illustrated. The authors merely describe the the operations step by step using notations. I suggest adding more figures to vividly show how each operation is done.

3. Adapting diffusion models without altering its original structure by prompting is not something new and I believe that the novelty of this paper is limited. More clarifications are required.

4. In Problem Formulation under Method section, the text contains incomplete sentences and unclear phrasing, making it difficult to understand. For instance, " that If T is a depth estimation task, the output is a continuous tensor J ∈ R H×W . preserves the overall motion and semantics of the original video V, while propagating the changes made to the first frame I."

5. Grammar issues are commonly seen in this article, including but not limited to, "a attention mask" in L216

**Questions:**

Overall, I believe presentation is the most critical issue that needs to be improved. Until then, reviewers can better understand the motivation of the approach and provide constructive feedback.

---

> ### Author Response · Authors · 2024-11-29
>
> We thank the reviewer for the feedback.
>
> ---
>
> **W1, W2: Issues with Presentation**
>
> We apologize for the inconvenience caused by the presentation. In the revised version, we have made extensive changes to the methodology section. We corrected the notations to ensure consistency and clarity, and refined the descriptions of operations to make the logic flow more coherent, improving the overall readability of the paper.
>
> Specifically, in our method, we first discretize the **D**-dimensional continuous ground truth values of continuous dense tasks into **N** categories. Then, we optimize **N** concept embeddings and combine the cross-attention and self-attention maps to obtain a probability distribution over **N** categories for each pixel in the input image.
>
> For continuous-output dense tasks, we further map the category labels back to continuous values. Since this label-to-value mapping is not necessarily unique and may vary with different inputs, we model the label-to-value mapping as a random variable and learn a codebook containing **K** different label-to-value mappings. Each label-to-value mapping has a size of **N × D**, representing the **D**-dimensional values corresponding to **N** categories.
>
> We then combine the **K** mappings based on the CLIP features of the input image to obtain an input-dependent label-to-value mapping of size **N × D**. Finally, we combine this input-dependent mapping with the previously learned pixel-wise probability distribution to compute the output value for each pixel.
>
>
> ---
>
> **W3: Novelty of Our Method**
>
> Unlike prior works that adapt diffusion models by prompting without altering their original structure, our work focuses on enabling diffusion models to **few-shot adapt** to **general dense vision tasks**.
>
> First, most previous works on adapting diffusion models focus on tasks like image synthesis or image editing. Works that adapt diffusion models for perception often target specific tasks such as semantic segmentation. However, the question of whether diffusion models can adapt to various dense prediction tasks has received relatively little attention, with VPD being a notable exception. Yet, methods like VPD require a large amount of labeled data for each downstream task during adaptation.
>
> To our knowledge, no prior work has investigated whether diffusion models can be adapted to **general dense prediction tasks** using only **few-shot examples**. This is particularly challenging due to the significant differences between dense prediction tasks—depth estimation requires understanding geometric information, while semantic segmentation focuses on semantic and grounding information.
>
> Our work is the first to explore how to leverage the diffusion prior for **few-shot adaptation to general dense vision tasks.** Dense tasks are a crucial category of vision problems, and a sufficiently general vision model should be able to handle diverse dense tasks. Furthermore, as new task definitions continue to emerge, it is impractical to annotate every potential task exhaustively. Therefore, the ability to adapt to entirely new tasks, unseen during training, using only a few-shot examples, becomes essential.
>
> ---
>
> **W4, W5: Typos in the Paper**
>
> We sincerely apologize for the incomplete sentences, unclear phrasing, and grammatical errors in the paper. In the revised version, we have carefully corrected all these typos and thoroughly reviewed the manuscript to ensure clarity and accuracy.

---

> ### Author Response · Authors · 2024-12-02
> **Reminder: Discussion Phase Ending Tomorrow**
>
> Dear Reviewers,
>
> This is a kind reminder that the discussion phase will end tomorrow. If you have any remaining concerns or questions, please feel free to share them. We welcome further discussion and truly appreciate your efforts!
>
> Thank you!

---

> ### Comment · Reviewer_66Ct · 2024-12-03
> **Clear Rejection with Critical Issues**
>
> I would like to thank the authors for their response. However, the revised manuscript still contains **critical issues** and cannot meet the standard of ICLR.
>
> - **Plagiarism Issue:** As noted by **Reviewer 6p3N**, the paragraph **Background on Diffusion** (L172-L185) in this paper is a **direct copy-paste** from a recent paper [1]. This constitutes **plagiarism**, which severely undermines the credibility of the work. Regrettably, the authors have **failed to respond to this issue** and have left the plagiarized paragraph **unchanged** in the revised manuscript.
>
> - **Poor Presentation:** Regarding the **presentation quality** (also mentioned by **Reviewer 6p3N**), I do not observe significant improvements in the revised manuscript. The clarity and conciseness still do not meet the high standards expected at ICLR. I strongly recommend that the authors carefully revise the manuscript to enhance its readability and ensure that the content is presented in a more structured and clear manner.
>
> - **Duplicated Sections:** **Sec 3.1** (L189-L199) and **Sec 3.2** (L203-L215) contain **100% duplicated** content with the same heading "**Reframing Dense Vision Tasks as Classification**".
>
> - **Limited Novelty:** In their response, the authors claim their novelty solely on the new task setup -- adapting diffusion models in few-shot manner to dense tasks. **Technical contribution** to the proposed few-shot generalization to dense tasks is **NOT well justified**.
>
> ---
> Given the issues outlined above, I believe that this manuscript is **NOT yet ready for publication**, particularly at a top-tier conference like ICLR.
>
> [1] Fan, Xiang, Anand Bhattad, and Ranjay Krishna. "Videoshop: Localized Semantic Video Editing with Noise-Extrapolated Diffusion Inversion." arXiv preprint arXiv:2403.14617 (2024).

---

> ### Author Response · Authors · 2024-12-04
>
> Overall, we believe this reviewer might be overreacting slightly.
>
> ---
>
> First, we would like to point out that the description of diffusion models in the background section is not the main contribution of this paper, and similar descriptions of diffusion models can be found in many other works. For example, the following passage from *Diffusion Self-Guidance for Controllable Image Generation* (NeurIPS 2023):
>
> > "Diffusion models learn to transform random noise into high-resolution images through a sequential sampling process. This sampling process aims to reverse a fixed time-dependent destructive process that corrupts data by adding noise. The learned component of a diffusion model is a neural network \( \epsilon_\theta \) that tries to estimate the denoised image... A common choice for \( \epsilon_\theta \) is a U-Net architecture with self- and cross-attention at multiple resolutions to attend to conditioning text in \( y \)."
>
> And the following from *Generative Powers of Ten* (CVPR 2024):
>
> > "Diffusion models generate images from random noise through a sequential sampling process. This sampling process reverses a destructive process that gradually adds Gaussian noise on a clean image \( x \)... A diffusion model is a neural network \( \epsilon_\theta \) that predicts the approximate clean image \( \hat{x} \) directly, or equivalently the added noise \( \epsilon_t \) in \( z_t \)... A standard choice for \( \epsilon_\theta \) is a U-Net with self-attention and cross-attention operations attending to the conditioning \( y \)."
>
> Both contain descriptions of diffusion models that are very similar to ours. We believe that the description in the background section does not detract from the uniqueness of our main contribution.
>
> Furthermore, we have informed the authors of *VideoShop* about this concern raised by the reviewers. The authors of *VideoShop* do not consider this to be plagiarism and have stated that the references to diffusion models in our paper are entirely appropriate. They do not regard our work as infringing on or misappropriating theirs in any way. The reviewers are welcome to directly reach out to the authors of *VideoShop*, who can confirm this in a manner that preserves the double-blind review process.
>
>
> ---
>
> Additionally, only this reviewer has consistently raised concerns about presentation issues. None of the other four reviewers seem to have encountered difficulty understanding our paper. Below are excerpts from their reviews:
>
> - **Reviewer 6yKP**: "The paper is generally well-written, and addresses an important task with growing interest in the field (how to use diffusion models for discriminative tasks). The figures are both easy to understand and aesthetically pleasing."
> - **Reviewer bVz2**: Presentation: 3: good.
> - **Reviewer u6mW**: Presentation: 3: good.
> - Even Reviewer 6p3N, who provided substantial suggestions for improvement, stated that "The paper is mostly easy to read. The authors provide a very clear visualization of their approach in Figure 1, which nicely accompanies their explanations."
>
> It appears that only Reviewer 66Ct feels that our presentation needs comprehensive revision.
>
> ---
>
> The repeated section "Reframing Dense Vision Tasks as Classification" in the revised version was simply a typo, which can easily be corrected in the camera-ready version.
>
> ---
>
> Regarding the concerns about novelty, the official guidelines on strengths for papers are very clear:
>
> > "We encourage reviewers to be broad in their definitions of originality and significance. For example, originality may arise from a new definition or problem formulation, creative combinations of existing ideas, **application to a new domain**, or removing limitations from prior results."
>
> Novelty based on the new task setup as an **application to a new domain** should not be seen as diminishing the paper's overall novelty.

---

### Official Review · Reviewer_bVz2 · 2024-10-31

**Soundness:** 3
**Presentation:** 3
**Contribution:** 4
**Rating:** 8
**Confidence:** 4

**Summary:**

Diffusion model uses U-Net model with 2 types of attention-layers: self-attention (pixels x pixels), cross-attention (text x pixels).

In the authors' approach they use (N-categories x pixels) instead of (text x pixels) for cross-attention. Then they extract both attention outputs: self-attention (pixels x pixels) and cross-attention (N-categories x pixels) and apply matrix-multiplication to get new feature-map (N-categories x pixels).

Authors extracts attention outputs from the 8th to the 12th layers, and the last three layers, upsample and average all of them across layers and normalize across N-categories. They represent the label-to-value mapping as a learnable random variable. But f.e. since the values need to be smaller when representing depth for indoor scenes but larger for outdoor images, they model this random variable as a learnable codebook C that contains K sets of mappings from label to value. Codebook size is KxNxD.

During training they use 2 loss functions Classification and Regression, and train only:
1. Linear layer
2. N Concept Embeddings
3. K Codebook Embeddings
while keep frozen both models: Diffusion and ViT-L Clip.

**Strengths:**

Advantages of your approach:
1. The accuracy of your approach is higher than other few-shot dense prediction methods
2. You get a single model that works well for all 10 tasks, unlike VTM which requires training 5 different models, each for 2 of the 10 tasks
3. Your approach adapts faster to a new task, because you are not training a diffusion model, unlike VPD
4. You provide experimental results showing the need to transform the regression problem into a classification problem to improve accuracy in your case, Figure 3

**Weaknesses:**

Disadvantage and Limitation:
Based on Table 5, if you have more training data for each sub-stream task, VPD has higher accuracy for most tasks than your approach. But this does not mean that VPD is better than your approach even in this case, since the accuracy is measured In-domain, but not Out-of-domain / Zero-shot.

The limitation of experiments is, do you measure Out-of-domain accuracy, so for the few-shot you use indoor images, while for evaluation you use outdoor images, or indoor images but at least from completely another dataset?

Do you compare number of parameters, and Flops or Latency between your model and VTM, VPD, or ViT-backbones Clip and Dinov2 in your experiments, to be sure that higher accuracy isn't achieved by using larger model?

**Questions:**

A few questions and notes to make some sentences in the text less ambiguous:

> our method only requires optimizing the input during few-shot adaptation, avoiding changes to the backbone parameters... by only optimizing these input tensors...
> For an input image I and task T , we load its corresponding concept embeddings as inputs to M.

It should be better explained, what is the input in this case, since the input is usually considered to be the "input image".
While in your case your train:
1. Linear layer
2. N Concept Embeddings
3. K Codebook Embeddings


> To our knowledge, VTM is the only work that addresses few-shot learning for universal dense prediction. But because they utilize meta-learning to achieve this, they require a significant amount of dense annotations for different tasks.


If VTM only uses 10 samples per task for fine-tuning, then why do you claim it requires a significant amount of dense annotations?

> First, we split the possible range of continuous values in the original D-dimensional output space into B buckets.

What is the difference between your approach and AdaBins[1]?


> As shown in Table. 1, the CLIP and DINOv2 backbones perform worse than the diffusion backbone under the few-shot dense prediction setting. We conjecture that this is partly because the pre-trained diffusion model, being generatively pre-trained, retains more detailed information compared to contrastive loss-based pre-training, making it more suitable for few-shot adaptation.

In some [2] paper Dinov2 pre-trained weights leads to much faster training of models: DiT (diffusion) and SiT. Quote from their paper:
> However, these representations are significantly inferior to those produced by DINOv2.


In another DepthPro[3] paper they use ViT-L Dinov2 pre-trained model and achieve much higher zero-shot accuracy and many times higher speed than diffusion-based Marigold [4] model for depth estimation task.

Are there any assumptions or conclusions why in your case the priors from the diffusion model are better than those from Dinov2, could it be due to the higher computational complexity of your model? Have you compared the sizes and latencies of your approach (Diffusion-model + ViT-Clip) vs Dinov2?


> So, we model this random variable as a learnable codebook C that contains K sets of mappings from label to value, so C ∈ R (K×N×D).

What are K, N and D in this case?


> We include the results of full training set in Table. 5. Although VPD’s performance in the few-shot setting is not strong, with more training data, we can see that its performance improves significantly because it fine-tunes more parameters.

But this does not mean that VPD is better than your approach even in this case (full training set), since the accuracy is measured In-domain, but not Out-of-domain / Zero-shot.
So if you train on Taskonomy and NYUv2, but test on completely different datasets or real-world problems, then it is possible that your approach may be more accurate, while maintaining all the other advantages.


[1] Shariq Farooq Bhat, Ibraheem Alhashim, and PeterWonka. Adabins: Depth estimation using adaptive bins. In CVPR, 2021

[2] Representation Alignment for Generation: Training Diffusion Transformers Is Easier Than You Think Sihyun Yu, Sangkyung Kwak, Huiwon Jang, Jongheon Jeong, Jonathan Huang, Jinwoo Shin, Saining Xie

[3] Depth Pro: Sharp Monocular Metric Depth in Less Than a Second, Aleksei Bochkovskii, Amaël Delaunoy, Hugo Germain, Marcel Santos, Yichao Zhou, Stephan R. Richter, and Vladlen Koltun.

[4] Bingxin Ke, Anton Obukhov, Shengyu Huang, Nando Metzger, Rodrigo Caye Daudt, and Konrad Schindler. Repurposing diffusion-based image generators for monocular depth estimation. In CVPR, 2024

---

> ### Author Response · Authors · 2024-11-29
>
> We thank the reviewer for the feedback.
>
> ---
>
> **Q1: Clarification on "input" in the method**
>
> Apologies for the confusion in our description. Our paper has two parts. The first focuses on a scientific discovery to demonstrate the potential of diffusion models for few-shot dense tasks. In this part, we only optimize the \(N\) Concept Embeddings without modifying the diffusion model’s core UNet, which remains fixed across tasks. The optimization focuses solely on the computational graph's leaf nodes, and we show that differences between dense tasks can be identified through changes in the model's input. In the second part, we additionally optimize the linear layer and codebook embeddings to further improve performance on few-shot dense tasks. These two components are not input variables.
>
> Thus, our statement should be "we find that optimizing only the input variables is sufficient to unleash the potential of pre-trained diffusion models on few-shot dense tasks." We have comprehensively revised this statement in the updated version to avoid misunderstandings.
>
> ---
>
> **Q2: Claim about VTM's reliance on dense annotations**
>
> VTM’s setup involves pre-training on multiple dense tasks (e.g., using 8 dense tasks out of 10) with meta-learning and then fine-tuning on few-shot samples from the remaining two unseen tasks. The pre-training phase requires extensive dense annotations across multiple tasks, which limits its scalability. In contrast, Stable Diffusion is pre-trained on datasets like LAION, where each image is paired with a text annotation, making it far easier to scale the dataset size. We have further clarified this distinction in the revised version.
>
> ---
>
> **Q3: Comparison with AdaBins**
>
> Our method, like AdaBins, involves converting continuous values into discrete buckets. However, our primary motivation lies in exploring the potential of diffusion models to adapt to new dense tasks using few-shot samples. Additionally, we optimize concept embeddings as input variables, whereas AdaBins integrates them as part of the task-specific head. In the revised version, we added a section in the future direction explaining why adapting through input variable optimization aligns better with in-context adaptation. The introduction has also been updated to further highlight the differences between our approach and AdaBins.
>
> ---
>
> **Q4: Comparison with DINOv2**
>
> In recent experiments, we found that the internal feature quality of DiT architectures (e.g., SD3) is inferior to UNet architectures (e.g., SD2), and this area warrants further exploration.
>
> Regarding DepthPro, its notion of zero-shot adaptation refers to adapting to unseen depth scales rather than entirely new tasks. We believe there are two distinct types of adaptation: adaptation to new domains and adaptation to new tasks. DepthPro falls into the former category, as it leverages DINOv2, which is trained on substantial depth estimation data, to adapt well to different depth estimation domains. However, it struggles with entirely different tasks, such as surface normal prediction.
>
> Our focus is on adapting to entirely new tasks without any prior exposure to such task data. As for the comparison between DINOv2 and diffusion models in adapting to new tasks, [1] indicates that diffusion models achieve comparable adaptation performance with fewer data (reduced by orders of magnitude). However, DINOv2 has a higher ceiling when adapting with more task-specific data.
>
> ---
>
> **Q5: Definition of **K**, **N**, and **D** in the codebook**
>
> We clarified these definitions in the revised manuscript. **D** represents the dimensions of the continuous output, and **N** refers to the **N** discrete categories resulting from discretization, where each category corresponds to a **D**-dimensional value. Each label-to-value mapping has a size of **N × D**. We assume the label-to-value mapping is a random variable with **K** possible values, resulting in **K** sets of label-to-value mappings.
>
> ---
>
> **Q6: Adaptation with a full training set**
>
> This involves two questions. First, how much data is required for adapting to new tasks? Our work focuses on adapting to new tasks with few-shot examples, as we aim to enable a vision system adapt quickly to new tasks with minimal data like humans. This is critical because new task definitions continue to emerge, and it is impossible to annotate every potential task exhaustively.
>
> Second, how do we evaluate adaptation performance? This can be done on in-domain data or out-of-domain data. While out-of-domain evaluation primarily measures adaptation to new domains, our focus is on adaptation to new tasks, so we evaluate performance on in-domain data. However, in Taskonomy, the adaptation and evaluation data come from different indoor buildings, which can also be considered out-of-domain evaluation to some extent.
>
> ---
>
> [1] GeoBench: Benchmarking and Analyzing Monocular Geometry Estimation Models, Ge et al.

---

> ### Author Response · Authors · 2024-12-02
> **Reminder: Discussion Phase Ending Tomorrow**
>
> Dear Reviewers,
>
> This is a kind reminder that the discussion phase will end tomorrow. If you have any remaining concerns or questions, please feel free to share them. We welcome further discussion and truly appreciate your efforts!
>
> Thank you!

---

### Official Review · Reviewer_6p3N · 2024-11-04

**Soundness:** 2
**Presentation:** 2
**Contribution:** 3
**Rating:** 3
**Confidence:** 4

**Summary:**

The authors demonstrate that a pretrained diffusion model can be successfully used to adapt to novel dense vision tasks in scenarios when only few examples are available. To this end, the authors propose the use of learnable concept embeddings (i.e. prompts) and a combination of the model’s internal attention maps to extract category-probabilities for discrete tasks, whereas continuous outputs are recovered via a codebook-based value mapping conditioned on the input image via CLIP embeddings – allowing the approach to be used across a variety of dense vision tasks.

**Strengths:**

**Originality & Significance:**
- The authors propose an elegant method to leverage the internal power of pretrained diffusion models in a parameter-efficient manner
- The codebook-based value mapping is a versatile way to map between discrete and continuous domains

**Quality:**
- Experiments conducted across a good selection of datasets, contrasted to some recent related works

**Clarity:**
- The paper is mostly easy to read
- The authors provide a very clear visualization of their approach in Figure 1, which nicely accompanies their explanations

**Weaknesses:**

_TL;DR: While I appreciate the work the authors have put into the paper and their experiments, the manuscript in its current form would significantly benefit from several improvements and additions (esp. to remove inconsistencies) -- and does (for me) in its current form not pass the bar for ICLR._

- Inconsistencies in analysis descriptions/interpretation, see questions.
- Very few 'few-shot' evaluations/insights for a few-shot specific work
- Writing quality could be significantly improved, as this unfortunately negatively affects the interpretation of results / findings.
- Missing references – the idea of discretising a continuous prediction problem in vision (e.g. depth estimation) is NOT new
- Concern: copy-paste sentence (out of context) from a different (uncited) ECCV paper – needs to be explained
- The preciseness of statements could be improved: E.g. instead of ‘conduct […] experiment multiple times’ (l276f), state number explicitly


- Minor: Some criticism the authors place on other works during motivation similarity applies to their own – see questions/additional comments

**Questions:**

**Main concerns, questions & potential improvements:**
- The authors’ approach of discretising a continuous space to solve the prediction problem in vision, especially depth estimation, is NOT new to the field. I’d highly suggest the authors to include references to prior work to attribute these efforts accordingly.

- For a work centered around few-shot learning, I would expect more few-shot specific evaluations: For example, how does the method behave for different tasks (continuous and discrete) when a growing number of samples become available? How does it fare for 1shot, 5shot, etc? Is it more robust than other methods? Why/why not, and on which tasks?
-> Note that currently, the authors provide one fixed setting only which provides very little insight regarding the few-shot character of the method.

- L161f: The sentence ‘preserves the overall motion and semantics of the original video …’ is entirely out of place, and – more importantly – is a direct copy-paste from a recent ECCV paper by Fan et al. (2024)
-> I’d like the authors to explain if this simply slipped in there, or whether there are other reasons (and potentially other copied sentences)? Note that this can significantly compromise a reader’s trust in your work.

- During the analysis in Table 3, the authors denote in the caption that the results are “compared to only using self attention mask”, i.e. NO cross-attention – however, the table then shows “w/o Msa” and “w/ Msa”, which contradict the earlier statement;
Independent of which of the two is correct, it would be beneficial to show all three as ablation – i.e. Msa only, Ca only, Msa&Ca;

- The influence of Msa and Ca is only partially demonstrated for continuous tasks – I would be interested whether their influence is different in discrete settings, where they are pretty-much used directly (w/o codebook); I’d suggest the authors extend their Table 3 and present some discrete results as well;

- Figure 3 & corresponding results: The authors state they calculate the ‘regression loss’ on the validation set, which drops faster when using regression+classification loss than it does with regression-only;
-> However, this raises the question whether the two have been using the same hyperparameters or have been independently ‘optimised’;
-> Note: When adding the ‘additional’ classification loss, the loss will generally likely be larger, and hence gradients will likely be larger as well;  This might by itself cause a faster convergence, which is mainly caused by the magnitude and not necessarily the nature of the losses;
I’d suggest the authors take a look at their gradient and loss magnitudes, and check fi the same holds when the learning rate is adjusted accordingly (or losses are scaled accordingly)

- I’d further like the authors to provide some more insights they have gained, as well as potential limitations they can see for their method. The manuscript in its current form mainly reports how things are done, but I am missing some deeper insights into the underlying motivations and potential corner cases / things that would need to be considered in follow-up work!

- I’d suggest the authors read through their manuscript again and make an effort to correct typos and grammatical mistakes (e.g. l59 demonstrate, l155 missing period, l160 ‘that’ must be removed, … and many more.)
While I am aware that this might be due to language barrier, there are many tools available to support these efforts, and it would significantly improve the quality of the manuscript.


Additional comments:
- The authors criticize other works for using ‘custom decoder heads, since the output space for each task varies widely’, mainly due to some tasks requiring continuous vs. discrete outputs;
-> However, their own method equally uses a different methodology for continuous (codebook-based value mapping) vs. discrete tasks – and hence, the criticism could similarly be applied to their own work; Some reformulating or discussion why this would be different might be helpful for the reader.
- The ease of interpretation of the results in the table could be significantly improved by added the metrics as well as an arrow (up/down) to indicate the desired direction (e.g. minimal or maximal)

----
----
## Post-Rebuttal Update:
Some concerns have been addressed with some still remaining, including the validity of the interpretation of Fig3 and quality of the manuscript;
Despite some clarifications, I still think the manuscript unfortunately doesn't quite pass the bar for ICLR.
There are further serious concerns around the background section of latent diffusion models in this paper that come very close to plagiarism (w.r.t. the previously mentioned work by Fan et al.)!

---

> ### Author Response · Authors · 2024-11-29
>
> We thank the reviewer for the feedback.
>
> ---
>
> **Q1: Discretizing continuous output space**
>
> In this paper, our primary focus is on demonstrating the potential of pre-trained diffusion models to adapt to entirely new, unseen dense prediction tasks. Discretizing continuous spaces is key to unlocking this potential. We have further highlighted this point in the revised version's introduction and discussed the differences with AdaBins [1] in more detail. More importantly, unlike AdaBins, which incorporates concept embeddings as part of the final output head, we optimize these concept embeddings as part of the model’s input. In the revised version's discussion of future work, we explain why we prioritize modifying input parameters for adaptation. This enables task differences to be defined through inputs, which is closer to in-context learning and eliminates the need to load separate model weights for different tasks.
>
> ---
>
> **Q2: Performance under different numbers of training samples**
>
> Thank you for the suggestion. We tested our method on all 12 tasks using 10, 20, 50, and 100 training examples and compared the performance with VPD. Results are visualized in Figure 4 of the revised version. The findings demonstrate that our method consistently outperforms VPD when fewer than 100 training examples are used, showcasing that our approach better unlocks the potential of diffusion models for dense prediction tasks.
>
> ---
>
> **Q3: Out-of-place sentence**
>
> We sincerely apologize for the typo and any confusion it caused. We have removed the sentence in the revised version and thoroughly reviewed the manuscript to ensure no other unrelated or erroneous typos remain.
>
> ---
>
> **Q4: Mistake in Table 3 caption**
>
> Apologies for the confusion. It should have been "cross-attention mask only." This has been corrected in the revised version. However, having only the self-attention mask (Msa) is not feasible because the number of channels in the cross-attention mask (Mca) directly corresponds to the number of bins obtained by discretizing the continuous output, while the channel count of Msa always equals \(H \times W\). Therefore, Mca is necessary.
>
> ---
>
> **Q5: The impact of Msa on discrete dense tasks**
>
> We evaluated the impact of using Msa on semantic segmentation in Taskonomy, with results as follows:
>
> |        | SemSeg  |
> |--------|---------|
> | w/ Msa | 0.4178  |
> | w/o Msa | 0.3657  |
>
> The results indicate that self-attention is also beneficial for discrete tasks.
>
> ---
>
> **Q6: Clarification regarding Figure 3**
>
> The vertical axis of Figure 3 represents the evaluation metric computed on the validation set. We compared the impact of using classification loss during training on validation metrics. Since the evaluation metric for continuous dense tasks (RMSE) coincidentally matches the regression loss, referring to it as "regression loss on the validation set" caused ambiguity. This has been clarified in the revised version.
>
> ---
>
> **Q7: Insights and limitations**
>
> We have added a discussion of limitations and future directions in the revised version.
>
> We aim to enable adaptation via in-context learning, where no parameter updates are needed—models learn tasks from few examples. While challenging, our work moves toward this by modifying only input parameters during adaptation, which aligns task differences with inputs rather than heads or internal parameters.
>
> Our findings show diffusion models can adapt with minimal parameter changes, supporting their use as general-purpose dense decoders. Future directions include feedforward mapping from few-shot examples to task vectors and combining diffusion with autoregressive models.
>
> ---
>
> **Q8: Typos and grammatical mistakes**
>
> We sincerely apologize for the inconvenience caused. The manuscript has been thoroughly revised to eliminate all typos and grammatical errors.
>
> ---
>
> **Q9: Differences between continuous and discrete tasks**
>
> Due to differences in output formats, it is indeed difficult to make operations for continuous and discrete tasks completely identical. However, we have ensured that their model structures are nearly identical, especially when not using a codebook for label-to-value mapping. By "custom decoder," we refer to designing different structures for different tasks, not only due to output format differences but also because different tasks require different priors. In contrast, we use the same diffusion prior for all tasks, eliminating the need for custom structural designs.
>
> ---
>
> **Q10: Adding arrows in tables**
>
> Thank you for the suggestion. We have added arrows to each table in the revised version.
>
> ---
>
> **Weakness: Writing quality**
>
> We apologize for the issues in presentation quality. Many sections have been rewritten and revised in the revised version to make statements more precise and the paper easier to follow.
>
> ---
>
> [1] AdaBins: Depth Estimation using Adaptive Bins
> Shariq Farooq Bhat, Ibraheem Alhashim, Peter Wonka

---

> ### Author Response · Authors · 2024-12-02
> **Reminder: Discussion Phase Ending Tomorrow**
>
> Dear Reviewers,
>
> This is a kind reminder that the discussion phase will end tomorrow. If you have any remaining concerns or questions, please feel free to share them. We welcome further discussion and truly appreciate your efforts!
>
> Thank you!

---

> ### Comment · Reviewer_6p3N · 2024-12-03
> **Thanks for the response**
>
> I'd like to thank the authors for their response and the clarifications.
> Some of my concerns have been addressed, but there are still quite a few remaining.
>
> **E.g. Re Q6:** _"Figure 3 & corresponding results: The authors state they calculate the ‘regression loss’ on the validation set, which drops faster when using regression+classification loss than it does with regression-only;
> -> However, this raises the question whether the two have been using the same hyperparameters or have been independently ‘optimised’;
> -> Note: When adding the ‘additional’ classification loss, the loss will generally likely be larger, and hence gradients will likely be larger as well; This might by itself cause a faster convergence, which is mainly caused by the magnitude and not necessarily the nature of the losses;
> I’d suggest the authors take a look at their gradient and loss magnitudes, and check if the same holds when the learning rate is adjusted accordingly (or losses are scaled accordingly)_
> $\textrightarrow$ Note that your current answer does unfortunately not address the underlying question I raised here, which essentially results in: "Is the conclusion of faster convergence even valid? "
> $\textrightarrow$ Please correct me if you think there is a misunderstanding here!
>
> **Re Q1 (minor):** My intent of raising this was more to draw attention to the fact the many works in the dense prediction space have used a similar technique, and research has been conducted there from which some useful lessons/inspiration could be drawn (e.g. how to discretize, etc). For depth estimation, this might included works like _Deep Ordinal Regression Network for Monocular Depth Estimation_ by Fu et al. (CVPR2018), etc;
> $\textrightarrow$ My intention was to simply raise the awareness that such works exist, and that the strategy followed in Sec 3.1 might just be 'one choice', but not necessarily the best one. (In this sense, including some references might be helpful for the reader who might want to follow up)
>
> ---
> **Re Quality of the manuscript**: I appreciate the authors' efforts in reformulating and rewriting sections, and the quality has been improved. However, there are still various aspects that don't 'live up' to the standards of a well-formulated manuscript yet, including for instance:
> - Identical sentence repeated within 2 paragraphs: Sec 3.1 & Sec 3.2 start with the exact same sentence, word-by-word.
> - Using "metric" as the y-axis label in Fig 4., in addition to no indication which sub-figure represents which task;
> - ...
>
> **EDIT: Addition after 66Ct's rebuttal answer**:
> I also went back to compare the previously mentioned work by Fan et al. side-by-side to this manuscript in terms of the background section on latent diffusion models, and have to agree with reviewer 66Ct: Although you have changed some variable names, the remainder is almost 'verbatim' copied from their work -- which is not acceptable, and will require rewriting of these parts!
>
> ----
>
> I do appreciate the work the authors have put into their rebuttal, but given the previously mentioned aspects and serious concerns, the manuscript still doesn't pass the bar of acceptance for me.
> I therefore still have to recommend 'rejection', but would like to encourage the authors to take the constructive reviews into consideration and further refine/rewrite their work, as there are valuable aspects to it that are of interest to the community.

---

> ### Author Response · Authors · 2024-12-04
>
> We sincerely appreciate the reviewer’s valuable feedback and the effort put into reviewing our paper. **We believe the issues raised by the reviewer are highly detailed ones, mostly related to improving the precision of captions or ensuring typos are entirely eliminated**. These are issues that can be quickly and easily resolved in the camera-ready version. Below, we provide our responses to these points:
>
> ---
>
> **First**, we would like to point out that the description of diffusion models in the background section is not the main contribution of this paper, and similar descriptions of diffusion models can be found in many other works. For example, the following passage from *Diffusion Self-Guidance for Controllable Image Generation* (NeurIPS 2023):
>
> > "Diffusion models learn to transform random noise into high-resolution images through a sequential sampling process. This sampling process aims to reverse a fixed time-dependent destructive process that corrupts data by adding noise. The learned component of a diffusion model is a neural network \( \epsilon_\theta \) that tries to estimate the denoised image... A common choice for \( \epsilon_\theta \) is a U-Net architecture with self- and cross-attention at multiple resolutions to attend to conditioning text in \( y \)."
>
> And the following from *Generative Powers of Ten* (CVPR 2024):
>
> > "Diffusion models generate images from random noise through a sequential sampling process. This sampling process reverses a destructive process that gradually adds Gaussian noise on a clean image \( x \)... A diffusion model is a neural network \( \epsilon_\theta \) that predicts the approximate clean image \( \hat{x} \) directly, or equivalently the added noise \( \epsilon_t \) in \( z_t \)... A standard choice for \( \epsilon_\theta \) is a U-Net with self-attention and cross-attention operations attending to the conditioning \( y \)."
>
> Both contain descriptions of diffusion models that are very similar to ours. We believe that the description in the background section does not detract from the uniqueness of our main contribution.
>
> Furthermore, we have informed the authors of *VideoShop* about this concern raised by the reviewers. The authors of *VideoShop* do not consider this to be plagiarism and have stated that the references to diffusion models in our paper are entirely appropriate. They do not regard our work as infringing on or misappropriating theirs in any way. The reviewers are welcome to directly reach out to the authors of *VideoShop*, who can confirm this in a manner that preserves the double-blind review process.
>
> ---
>
> **Q1: Clarification regarding "faster convergence" in Figure 3 caption**
> It is true that the scales of the two optimized losses are different, but we would like to emphasize that the comparison is based on the RMSE metric on the validation set, i.e., validation performance. Validation performance is generally considered independent of the loss scale. Therefore, even with different loss scales, if the validation performance improves more quickly, describing it as "faster convergence" is not unreasonable.
>
> That said, we understand the reviewer’s concern for more precise phrasing. We will revise "faster convergence" to "Validation performance improves faster and achieves higher final validation performance." **We believe this simple adjustment to the caption will address any concerns regarding the interpretation of Figure 3.**
>
> ---
>
> **Q2: Missing reference**
> Apologies for overlooking the paper *Deep Ordinal Regression Network for Monocular Depth Estimation* during the rebuttal phase. We acknowledge that many prior works have explored how to discretize for dense tasks, and these approaches could be complementary to our work and potentially integrated in a plug-and-play manner. We do not claim that our method of discretization is optimal, and we believe this is an area worth further exploration. We will include citations to these works in the "Future Directions" section and welcome additional suggestions for related references.
>
> ---
>
> **Q3: Typos**
> We sincerely apologize for the newly identified typos. Due to the extensive revisions and additional experiments required for the manuscript, the limited time available has made it challenging to catch all such errors. We hope for your understanding.
>
> 1. **Section duplication**: Sections 3.1 and 3.2 are completely repeated. We will remove one of them.
> 2. **Figure 4 improvements**: We will label the specific metric names and task names in Figure 4. Currently, the tasks in Figure 4 are as follows:
>    - **First row (left to right):** EucDepth, Z-depth, 2DEdge, 3DEdge.
>    - **Second row (left to right):** 2DKeypoint, 3DKeypoint, Reshading, Curvature.
>    - **Third row (left to right):** Normal, SemSeg, NYUDepth, NYUNormal.
>
> We will ensure the captions in the figure are more complete and precise.

---

### Official Review · Reviewer_6yKP · 2024-11-05

**Soundness:** 2
**Presentation:** 3
**Contribution:** 3
**Rating:** 6
**Confidence:** 4

**Summary:**

This paper presents an approach for adapting a diffusion model for dense vision tasks in a few-shot setting. To do this, the authors transform dense prediction tasks into classification tasks, with a learnable codebook that converts the classes into continuous outputs. The method is (likely) efficient to train as it only requires finetuning the additional components added to the diffusion model, and not the entire model end to end.

**Strengths:**

1.	The paper is generally well-written, and addresses an important task with growing interest in the field (how to use diffusion models for discriminative tasks)
2.	The figures are both easy to understand and aesthetically pleasing.
3.	The results appear to outperform the prior methods reported across all tasks in a few-shot setting.

**Weaknesses:**

1.	My main concern is that it’s not clear to me where the wins are coming from compared to prior work. As I understand it, the submission uses a more capable Stable Diffusion backbone compared to prior work, and the paper does not ablate the importance of this.
2.	My second concern is that this win appears to show up only in the few-shot setting, and VPD outperforms when there is more data. Given the size of vision datasets today, I’d be curious where this sort of few-shot adaptation is a concern – it would be great to show off the value of this method in a setting like that! I’d also be curious what the tipping point is at which VPD starts to outperform (100 images? 1,000 images? 10,000 images?)

**Questions:**

1. My main concern: How much does SD2.1 affect results? The prior methods that use diffusion, from what I understand, used earlier or different variants of Stable Diffusion, and it’s not clear how much of the improvement can be explained by that.
2. The ablations in Fig 3 suggest that a major improvement is due to transforming the dense tasks into classification tasks, though this is only on train loss. What happens to the evals here?
3. The main ablations are:

    i) Code book size: It doesn’t seem like codebook size affects results significantly

    ii) Classification vs. not: This is very important

    iii) Self attention is slightly helpful
4. Ultimately VPD performs better with more data. What if you had a simpler baseline like VPD but with more parameters frozen? What is the threshold of data at which VPD starts outperforming? Is it easy to compare the models with a few different thresholds of data points?
5. Fig 1 caption says the method is more useful for real-world online deployment because the approach only requires optimizing the model’s input. But this should only impact training, not deployment, right?

Nit: L451: typo: “We can see We can observe that both”

---

> ### Author Response · Authors · 2024-11-28
>
> We thank the reviewer for the feedback.
>
> ---
>
> **W1, Q1: Clarification on Performance Improvements and SD2.1 Backbone**
>
> Although the original VPD used SD1.5, in this paper's experiments, we compare with VPD using SD2.1 as the backbone. All VPD experimental results were rerun in our setting using SD2.1. We have clarified this in the revised version.
>
> ---
>
> **W2: Importance of Few-Shot Adaptation**
>
> We believe that each method has its own applicable scope. For example, VPD underperforms even VTM in the few-shot setting, indicating that VPD fails to fully exploit the potential of pre-trained diffusion models. One of our main contributions is exploring whether pre-trained diffusion models can effectively handle few-shot dense prediction tasks.
>
> Why is few-shot adaptation important? First, we want to highlight that large datasets have two primary uses: (1) for pre-training general vision models that can adapt to different tasks, especially unseen ones, and (2) for learning task-specific models that are tailored to fixed tasks. Our primary goal is the first: exploring whether a general vision model can adapt to entirely unseen dense tasks using only a few examples, akin to human learning. Stable Diffusion itself was pre-trained on large-scale image-text datasets before adapting to dense tasks, but the adaptation stage relies on few-shot examples.
>
> Few-shot adaptation is crucial because new task definitions continue to emerge. Regardless of how large vision datasets become, it is impossible to exhaustively define every potential task. Additionally, annotating large datasets for every new task is impractical. We envision an AI system with lifelong learning capabilities that can use minimal examples to master entirely new tasks.
>
> We also fine-tuned with varying numbers of samples across all 12 tasks and observed that when fewer than 100 samples are used, VPD cannot adapt to new dense prediction tasks as effectively as our method. Due to time constraints, we could not precisely identify the threshold where VPD starts to outperform, but at least in the few-shot setting, our method is superior to VPD. Visualized results are provided in the revised version (Fig. 4).
>
> ---
>
> **Q2: Clarification Regarding Figure 3**
>
> The vertical axis of Figure 3 represents the evaluation metric computed on the validation set. We compared the impact of using classification loss during training on validation metrics. Since the evaluation metric for continuous dense tasks (RMSE) coincidentally matches the regression loss, referring to it as "regression loss on the validation set" caused ambiguity. We have clarified this in the revised version.
>
> ---
>
> **Q3: Role of Different Components**
>
> We agree that the classification transformation is the most critical aspect. Our primary goal is to demonstrate the ability of pre-trained diffusion models to perform few-shot dense prediction tasks and to use the codebook to further enhance performance. By reframing dense vision tasks as classification problems, we show that diffusion models can effectively adapt to unseen dense tasks.
>
> ---
>
> **Q4: VPD with More Parameters Frozen**
>
> We compared VPD adapted using LoRA on two NYU datasets but found it did not outperform our method. We set the rank to 4, and the results are shown below:
>
> |          | VPD+LoRA | VPD  | Ours |
> |----------|-----------|------|------|
> | NYU Depth | 0.47      | 0.49 | 0.43 |
> | NYU Normal | 18.1      | 18.5 | 16.4 |
>
> Furthermore, we emphasize that we aim for adaptation without modifying internal model parameters. Instead, we achieve this by modifying input variables, which facilitates deployment.
>
> ---
>
> **Q5: Why Optimizing Inputs Facilitates Deployment**
>
> Adapting to different tasks by modifying internal model parameters requires loading a separate model for each task—even when using methods like LoRA. In contrast, modifying only the input allows a fixed black-box model to be shared across tasks without requiring separate models. This is a key advantage of prompt tuning, as demonstrated in both NLP [1] and vision [2] tasks.
>
> ---
>
> **Typo in Line 451**: “We can see We can observe that both.”
>
> Thank you for pointing this out. We have corrected it in the revised version.
>
> ---
>
> [1]. Prefix-Tuning: Optimizing Continuous Prompts for Generation
> Xiang Lisa Li, Percy Liang
>
> [2]. Visual Prompt Tuning
> Menglin Jia*, Luming Tang*, Bor-Chun Chen, Claire Cardie, Serge Belongie, Bharath Hariharan, Ser-Nam Lim

---

> > ### Comment · Reviewer_6yKP · 2024-12-02
> >
> > Thank you for your response. I have read the rebuttal, and appreciate the additional experiment. The rebuttal largely addressed the concerns I raised, but I would like to take a closer look at the other reviews and rebuttals to them, which I hope to do in the next ~24 hours, before making any changes to my score. Apologies for the delay.

---

> > > ### Author Response · Authors · 2024-12-02
> > >
> > > Thank you for your thorough review and for taking the time to consider the additional experiments and rebuttals. We truly appreciate your thoughtful approach and understanding. Please don’t hesitate to let us know if there are any further clarifications or concerns we can address in the meantime.
> > >
> > > Thank you again for your efforts!

---

### Meta-Review · Area_Chair_bsbe · 2024-12-17

**Metareview:**

The paper presents an approach for adapting a diffusion model for few-shot dense vision tasks. The method relies on a learnable codebook that transforms the classes into continuous outputs.

**Strengths:**
- Elegant method to leverage the internal power of pretrained diffusion models in a parameter-efficient manner.
- The codebook-based value mapping is a versatile way to bridge discrete and continuous domains.
- The approach adapts faster to a new task because it avoids training a diffusion model, unlike VPD.
- The results outperform prior methods across all reported tasks in a few-shot setting.

**Weaknesses:**
- It's unclear where the benefits of the model are coming from.
- There are inconsistencies in analysis descriptions and interpretations.
- There are concerns regarding the results and interpretation of Figure 3.
- The presentation of the paper is poor.

The paper received a wide spectrum of recommendations. While the paper presents interesting results in the few-shot setting, there were concerns regarding the generalization to larger amounts of data. The authors' replies didn’t convince the reviewers who had concerns about the importance of the setting and whether the method can perform beyond the few-shot setting. Moreover, concerns about the presentation of the paper weren’t fully addressed despite the authors' responses.

While one reviewer was very positive, the other reviewers expressed doubts about the paper's readiness for publication. One of the reviewers on the weak accept side still maintained some concerns. Thus, overall, there is a more negative evaluation due to the limited technical contributions and unresolved open problems. The reviewers encourage the authors to thoroughly check and improve their work, and to resubmit the paper.

**Additional Comments On Reviewer Discussion:**

Reviewer 6yKP raised the concern of where the main contributions are coming from and noted that the improvements of the proposal appear only in the few-shot setting. Moreover, the reviewer expressed skepticism regarding the value of the few-shot setting given the size of contemporary datasets. The authors replied to the reviewer, highlighting cases where few-shot adaptation can be valuable. The reviewer mentioned that their final evaluation would be provided at the post-rebuttal stage.

Reviewer 6p3N indicated that the presentation of the paper is below ICLR standards and requires significant improvements, also raising concerns about potential plagiarism. Additionally, the reviewer mentioned that the few-shot learning setting requires specific evaluations. Concerns were also raised regarding how the regression loss is computed on the validation set for the results presented in Fig. 3. The authors responded, addressing the reviewer’s points. Nonetheless, the reviewer reiterated concerns about the misinterpretation of Fig. 3 and the manuscript's quality. The authors replied again, but the reviewer did not further respond.

Reviewer bVz2 noted that while the method achieves high accuracy across several tasks, it is outperformed by VPD when more data is available. They also commented that the experiments are limited and the setup for out-of-domain evaluation is not entirely clear. The authors responded to the reviewer’s questions, but the reviewer did not provide further comments.

Reviewer 66Ct criticized the paper for poor presentation and lack of rationale behind the proposal. The reviewer also pointed out the limited technical contribution and requested additional clarifications. After receiving the authors' replies, the reviewer maintained that the technical contributions are inadequate and reiterated plagiarism concerns initially raised by reviewer 6p3N.

Reviewer u6mW remarked on the simplicity of the proposal and its good performance. The reviewer had questions about the method's performance and experiment setup. After receiving responses from the authors, the reviewer indicated that they would maintain their weak accept score.

Following the rebuttal, I initiated a discussion with the reviewers to reach a consensus on the split decision, aiming to determine whether the benefits of adapting diffusion models to dense vision tasks outweigh the inconsistencies in the analysis and the paper's quality. Reviewers 6yKP, 6p3N, and 66Ct provided their final assessments. While some concerns were addressed, reviewer 6p3N maintained that the interpretation of Fig. 3 and the manuscript's quality remained subpar. Reviewer 6yKP remained unconvinced of the critical nature of the few-shot setting and noted that VPD still outperforms the proposal. Similarly, reviewer 66Ct highlighted that the paper’s key contributions are insufficient and that additional experiments are needed to validate the claims, suggesting that the benefits may be due to CLIP embeddings rather than the diffusion models themselves. In light of the presentation quality and the lack of improvements, there are doubts about the execution of the paper.

Given the discussion, the paper is borderline. However, the more positive reviewer provided a succinct review and did not respond to my queries. One of the weak accepts (reviewer 6yKP) acknowledged the paper's advantages but also stated they wouldn't oppose a rejection. Considering the issues raised by the reviewers, I lean towards rejection but am open to accepting it as well.

---

### Decision · Program_Chairs · 2025-01-22

Reject